# Public and Patient Involvement in Migration Health Research: Eritrean and Syrian Refugees’ and Asylum Seekers’ Views in Switzerland

**DOI:** 10.3390/healthcare12161654

**Published:** 2024-08-20

**Authors:** Afona Chernet, Daniel H. Paris, Lujain Alchalabi, Jürg Utzinger, Elisabeth Reus

**Affiliations:** 1Swiss Tropical and Public Health Institute, Kreuzstrasse 2, CH-4123 Allschwil, Switzerland; daniel.paris@swisstph.ch (D.H.P.); lujain.alchalabi@swisstph.ch (L.A.); juerg.utzinger@swisstph.ch (J.U.); elisabeth.reus@swisstph.ch (E.R.); 2University of Basel, P.O. Box, CH-4003 Basel, Switzerland

**Keywords:** asylum seekers, engagement, Eritrea, migrants, migration health research, public and patient involvement, refugees, Switzerland, Syria

## Abstract

Prior research has highlighted important healthcare access and utilization issues among new forced immigrants. We aimed to explore the role that public and patient involvement (PPI) might play in enhancing accessibility and specific contributions to migration health studies. We conducted open and in-depth interactive virtual discussions with asylum seekers and refugees from Eritrea and Syria in Switzerland. The PPI establishment consisted of three phases: inception, training and contribution. Prior to training, the concept of PPI was not straightforward to grasp, as it was a new approach—however, after training and consecutive discussions, participants were ardent to engage actively. We conclude that PPI holds promise in regard to raising awareness, improving healthcare system accessibilities and utilization, and enhancing and strengthening migration health research. Indeed, PPI volunteers were keen to raise their community’s awareness through their networks and bridge an important gap between researchers and the public.

## 1. Introduction

### 1.1. Background and Context

Every year, Switzerland hosts a considerable number of immigrants, including forced immigrants (particularly refugees and asylum seekers) in addition to regular workforce immigrants, which is comparable to other European countries. A forced immigrant is subject to migration with an element of coercion, including movements of refugees and internally displaced persons as well as people displaced by disasters, famine, etc. Among a population of 8.7 million in 2023 [1], the Swiss State Secretariat for Migration (SEM) reported nearly 2.3 million permanent and non-permanent foreign residents living in Switzerland in 2022 [2].

The mounting figure of forced immigrants, asylum seekers and refugees raised concerns among the public, policy makers, social servants and healthcare service providers. There are important health disparities among the arriving populations when compared to resident communities, including infectious diseases, undiagnosed and untreated non-communicable diseases, mental health and mother-and-child health issues. The constant challenges regarding the healthcare provision of forced immigrants in their host countries, and the need to elevate awareness among public health experts and policy makers, highlight that dedicated research to generate improved evidence is much needed. There is a large number of research projects addressing the understanding of health problems and needs of newly arriving refugees and asylum seekers [3,4]. However, evidence regarding the contribution of migrants in health- and care-related research activities, in terms of engagement and involvement as co-producers in the life cycle of a research project, remains scarce.

In an interactive exchange and in-depth engagement with asylum seekers and refugees from Eritrea and Syria, we uncovered key perceptions, expectations and context-specific know-how on health research among migrants. We jointly explored how these perceptions and existing knowledge can contribute to enhancing the quality and integrity of migration health research.

### 1.2. Experiences from a Previous Research Project

In 2015–2016, a research project was implemented to assess the major healthcare disparities of newly arrived asylum seekers and refugees in Switzerland through screening and surveillance over a 12-month period [5]. The study highlighted important public health concerns among newly arrived immigrants, including a limited understanding of the country’s healthcare system, inefficient utilization of services and a lack of information on help-seeking behavior—especially for mental health issues [6]. Additionally, language barriers that are critical to successful integration were identified [7].

The perceived needs and expectations of asylum seekers are connected with economic capital, such as daily income [8]. Employment opportunities contribute substantially to the well-being of migrants [9]. However, asylum seekers in some countries seem to be unsatisfied with the job opportunities offered [10].

The level of understanding of how the healthcare systems in host countries function plays a key role in the resulting access and actual utilization. The majority of refugees and asylum seekers from resource-limited countries are confronted with numerous challenges, such as cultural, language and religious barriers, unfamiliarity with the system and an inability to cover the costs of services. The more these barriers are apparent, the less likely it is that forced immigrants will access the health system and its services, which might further widen the gap between the service providers and the patients.

### 1.3. Public and Patient Involvement

#### 1.3.1. General Considerations

To address these issues, a dedicated public and patient involvement (PPI) strategy was created for migration health research at the Swiss Tropical and Public Health Institute (Swiss TPH), an associated institute of the University of Basel. The migration PPI was founded on the concept of working directly with the public in identifying or co-creating adapted research questions and their potential solutions. PPI is defined as “research being carried out ‘*with*’ or ‘*by*’ members of the public, rather than ‘*to*’, ‘*about*’ or ‘*for*’ them” [11].

Through PPI group meetings, refugees and asylum seekers can learn how to efficiently access and benefit from available services and contribute to improving research by actively participating as key informants for their fellow migrants. Additionally, they can learn how to co-develop specific research projects by addressing critical questions and generating evidence for the most relevant issues that interest their community. Hence, participation and engagement through PPI can strengthen the credibility of the research conducted in those communities, as they say, “*nothing about us, without us*” [12].

#### 1.3.2. The Concept of PPI

PPI in clinical research is a fundamental pillar to improve patient-oriented healthcare services and healthcare system assessments. PPI plays a pivotal role in the interaction of key stakeholders in healthcare research activities, as it connects the population with patients and the healthcare system to researchers [13]. PPI plays a particularly important role in health research for the adult population as primary consumers of the healthcare system by covering the large individual diversity [14,15]. PPI in clinical research aims to promote a deeper understanding of all key stakeholders involved and to help prioritize evidence generation for problems or solutions for policy makers and other stakeholders to induce change (Figure 1).

The concept of PPI is not new (Appendix A); yet, only about 20% of the recent peer-reviewed literature in the field of migration health included PPI reports [16] with considerable geographic heterogeneity. In the People’s Republic of China, only 3.4% of the published work included PPI aspects, while a much higher percentage of 44.5% was found for the United Kingdom. Some countries have created international collaborations and exchange platforms to tackle these challenges and share PPI experiences [17]. In Switzerland, the implementation of a shared decision-making process (SDM) within a highly decentralized healthcare system is taking place, and the importance of PPI is being increasingly recognized—particularly in areas of clinical research, trials and as a methodology for medical school graduates [18]. Likewise, Swiss TPH researchers and their partners have practiced the concept of engaging the public in research for over 20 years now [19,20,21].

As an institute with an 80-year history of research, education and services in public and global health with a pointed emphasis on low- and middle-income countries (LMICs), understanding the cultural, social and traditional norms of the population before planning any research is unavoidable [22]. Following the Commission for Research Partnerships with Developing Countries (KFPE; https://kfpe.scnat.ch/en), all researchers start their discussions with stakeholders at the conception stage of research projects and set the agenda together during the planning and preparatory phases [23].

Multiple scenarios for PPI have been highlighted—not only concerning health care directly but also indirectly in cases such as land and water use, animal studies for One Health projects and mother-and-child health [24,25,26]. This requires a transdisciplinary approach, involving cross-cutting expertise bridging many disciplines, i.e., anthropology, data sciences, epidemiology, public health, sociology and veterinary medicine [27,28].

PPI practice in LMICs is linked to a paucity of publications on engaging patients and the public in clinical research [29]. Though PPI is useful and supports research at every stage of the research’s life cycle [30], it is most effective if implemented in the early stages of research projects to enhance the quality and appropriateness of population-based studies [31]. Co-designed and co-produced projects involving both the public and patients have significant effects on reducing unnecessary healthcare-related research wastage [32]. Moreover, PPI is not only supportive for senior researchers but is also pertinent to early career academics and professionals in regard to optimizing the value, integrity and quality of their research [33].

## 2. Methods

### 2.1. Exploring Migration PPI through an Interactive In-Depth Exchange: Health Research during the COVID-19 Pandemic

During the COVID-19 pandemic, when stringent public health measures were implemented in 2021, we felt an urgency to start a PPI group. Hence, we contacted forced immigrants (i.e., asylum seekers and refugees) to better engage with them and enable us to have a deeper understanding of the research process among these populations. Our aim was to enhance trust in forced immigrants towards contributing, engaging and actively participating in public health research. Because we created this platform, individuals were able to share their healthcare needs and priorities. As specific COVID-19 prevention and restriction measures were imposed, including prolonged lockdowns over parts of the year 2021, the need for a PPI group became even more apparent than initially anticipated. The PPI group consisted of seven adult males and seven adult females from Eritrea and Syria. Meetings were adapted and re-organized taking into consideration COVID-19 restrictions. The group mostly met virtually.

### 2.2. Selection of Volunteers

Participants of this PPI project were either enrolled from a previous study [5] or invited to participate through existing networks. The three main reasons for contacting people to take part in the new PPI project were as follows. First, they were familiar with their home or origin country’s healthcare services, including benefits and challenges. Second, as they arrived in Switzerland a while ago, they were somewhat acquainted and already partially integrated into the host country. Third, they had already settled in Switzerland and were part of the migrant network, hence representing the migrant population in the country.

### 2.3. Stages of Establishing a PPI for Migration Health Research

As represented in the flow chart depicted in Figure 2, the establishment of a novel PPI structure for migration health research consisted of three distinctive phases: (i) the inception phase; (ii) the training phase; and (iii) the implementation phase. Asylum seekers and refugees with no background in PPI knowledge and experiences were invited to take part in the current migration PPI project. Participation was voluntary, and no incentives were provided.

During the inception or conception phase, necessary information, such as the meaning of a PPI and the roles and responsibilities of volunteers, were introduced and explained. In the following training phase, continuous meetings and discussions were held virtually mainly due to COVID-19 measures. Volunteers had the opportunity to practice and express their views, opinions, suggestions and other valuable information for PPI as part of their training. In the implementation phase, volunteers were further involved in some existing health research projects or activities, such as serving as interpreters and translators.

From an early stage, every member of the PPI was motivated to share his or her opinion freely on any related topics. This subsequently fostered discussion and “*broke the ice*” to allow for opinions, expressions and experiences to be shared without any limits or boundaries of ideas, thoughts and feelings. Through these continuous meetings and discussions, group members would advance in their understanding about the PPI and raise important health concerns, which eventually could develop into a specific research question.

In the following stage, the focus of discussion topics was on “*What is health research?*”, “*How is it conducted?*”, “*Who is responsible?*” and “*What procedure does it follow?*”—these topics were jointly explored. The moderator assisted in guiding the discussion and motivated the participants to raise critical issues to engage within the group.

In the advanced stage, the PPI group provided active input on either ongoing research projects or projects under construction. Involving the PPI group at the early stage of a research project, from setting the research question and proposal writing stage, was proven effective in producing quality research project [30,34,35,36].

### 2.4. Realization of the PPI Establishment

Regular monthly meetings were often challenging due to the participants’ situations, such as work, language learning, family responsibilities and internet accessibility and connectivity. Finding a convenient time for all participants was crucial. Hence, meetings were arranged every 4–6 weeks. Table 1 summarizes the characteristics of the first meetings held among Eritrean and Syrian PPI volunteers.

### 2.5. Criteria for Evaluation, Including Quantitative and Qualitative Indicators

In order to evaluate the PPI migration health research group—the first of its kind in Switzerland—some key indicators were identified for its success.
*Participation rate and adherence:* At every virtual meeting, participation of more than 70% of the volunteers was regarded as good intention and will of the group for the success of the initiation.*Active participation and contribution:* Bringing new ideas, questions, comments and suggestions about their community was regarded as measure of gradual increase in the concept of the PPI and their motivation.*Inviting new volunteer compatriots:* This was considered a gauge for the increasing awareness of the importance and effect of PPI in health research.*Proposing new discussion topics:* This was an indication that volunteers and the way they reach out to their communities to explore healthcare-related disparities or problems in their communities were enhanced.

## 3. Results

### 3.1. Reaching Out to Participants

Volunteers were initially recruited through existing migrant centers and networks due to the COVID-19 pandemic. Overall, 14 volunteers participated (Table 1).

The findings or views regarding the know-how of asylum seekers and refugees on PPI and its role in health research activities are summarized in the following four thematic subtopics. The ideas, views, comments and suggestions made by the participants were funneled into tangible thematic presentations that served as evidence-based input.
*Introduction and explanation of PPI among migrants**Willingness to attend a PPI migration health research programme**Engagement and contribution of participants**Input and propositions of PPI members*

### 3.2. Introduction and Explanation of PPI among Migrants

The concept of PPI was initially hard to grasp for the volunteer participants, as it was new, and they were not used to openly discussing their health needs. Several questions and inquiries underscore this issue: “*What is the difference between PPI and participating in research?*”, “*What benefit does PPI have for migrants?*”, “*How does PPI work?*” and “*What if someone does not have a medical background or training as health professional?*”

The clarifications provided by the coordinators proved effective, as participants became more involved in the discussions. Basic information on how PPI works and benefits the public, particularly in research, was provided to participants with palpable examples.

### 3.3. Willingness to Attend a PPI Migration Health Research Programme

All participants expressed a high interest in the initiative. They considered the project an opportunity to openly share their know-how. The exposure to, and personal experiences with, research including challenging discussions with peers and instructors were much appreciated. This open platform for discussions and exchange uncovered obstacles that hindered access to the healthcare system and its effective utilization and was an opportunity to bridge science and the public for a deeper understanding of health and well-being of vulnerable populations. Forced immigrants, particularly those deprived or isolated from the healthcare system facilities and privileges, undocumented migrants and asylum seekers waiting to get a permit to stay in the host country, are predominantly exposed to the risk of isolation from the healthcare system. Hence, through PPI group representation of their community, their healthcare access barriers, health service needs and priorities could be shared with investigators and were considered in the subsequent research which led to improved information and evidence for policy makers and funding organizations.

### 3.4. Engagement and Contribution of Participants

Despite the diverse backgrounds of volunteer participants (which included non-healthcare related professionals), an active contribution to lively and informative discussions was provided by all. They abundantly shared personal experiences in healthcare services, be it in their home countries or while being refugees in the host country. Others shared experiences of their loved ones or explained what they picked up through mass media or social media. Discussions were interactive and highly engaged and shared a vast body of experiences, which provided valuable ideas for potential research topics in further studies.

### 3.5. Input and Propositions of PPI Members

Most inputs pertained to communities’ healthcare service challenges. Recurrent questions were asked, such as “*How can we overcome barriers to accessing healthcare services in Switzerland?*”, “*Why is the insurance system so difficult to understand?*”, “*Where can newly arrived asylum seekers and refugees get information about how the healthcare service in the country works easily?*” and “*How can culturally competent healthcare service be developed and implemented for migrants?*”

Additionally, specific inquiries were raised that require more prominent consideration in migration health research, concerning maternal and child health care, regular checkups, dental healthcare insurance, family physician and regulations. Others wondered about drug prescriptions such as antibiotics, as they noted considerable differences between their home or county of origin and the host country Switzerland.

## 4. Discussion

Migration remains a central and critically important topic of the century—its potential for both positive and negative impacts on individuals and societies as a whole needs to be better realized and appropriate methodological approaches employed to respond to the associated challenges—this is where PPI can play a crucial role [37]. Integrating PPI and participatory elements into migration, mobility and health research can enable bi-directional exchange between migrants and researchers and help develop a beneficial dialogue to co-create improved formulation of research questions, selection of appropriate methods for data collection, project study designs, interpretation of results and implementation or dissemination of research findings [38].

### 4.1. Role and Effect of PPI in Raising Awareness of Research Importance

In general, participants of this first PPI initiative in migration health research in Switzerland agreed that the initiative provided an important bridge to connect the community with academia. “*The migrant community needs to be actively engaged in health research, as it is for their benefit and the benefit of other communities. In addition, also research is conducted for the benefit of future generations. Hence, the community needs to consider and care for future generations by engaging themselves in research activities*” (37-year-old male migrant from Eritrea). Additionally, it was emphasized that “*Health research is a means of educating the community and provides an opportunity of gathering information for educating the public and raising awareness, and can positively impact the health of the people*” (37-year-old male migrant from Eritrea).

The following example highlights the personal experience of a PPI member from an earlier migration health study [5]. “*I benefited from partaking in a migration health screening research project. Asymptomatic parasitic diseases were diagnosed. Hence, self-involvements in health-related research projects benefits the participants both through access to medical care service and increasing awareness of the health system and related information*” (40-year-old male migrant from Eritrea).

Despite the anxiety forced immigrants may have, they are aware of their needs for early screening and subsequent management of specific health issues they have been identified. They are also aware of the importance of adequate treatment as quickly as possible according to what the healthcare system can offer. Nevertheless, due to misconceptions, mistrust, suspicions and fear of stigmatization, most of them hesitate to visit any facilities to seek help and join research activities.

“*It is obvious that there are symptomatic and asymptomatic health problems. We have difficulties and problems. We consider ourselves as healthy; however, we know that we are not. It was only after we were diagnosed and participated in research projects such as the migration health study, conducted by Swiss TPH, that we became aware of some health issues. If our participation in research is low, you (the researchers) need to raise our awareness*” (40-year-old male migrant from Eritrea).

Researchers not only need to raise awareness of migrants to partake in research studies, but they also need to understand the problem of the community, in order to raise appropriate research questions. “*Only the people themselves know best about their problems*” (38-year-old male migrant from Eritrea).

In order to identify the best solutions for migrants’ health problems, migrants themselves are able to provide valuable information and can contribute to the facilitation of problem solving. Hence, clinicians and researchers involved in migrant health need to critically assess whether they really involve, engage and empower the public and patients through consulting, partnering and authorizing [39].

From migrants’ perspective, one PPI member emphasized mental health care and support-related studies’ needs as follows: “*Most studies [medical research studies including mental health] focus on diseases instead of the causes for the diseases. Among the causes to be mentioned are worries and stress leading to different diseases. It is better to concentrate on the root cause for those worries, stressors and others*” (40-year-old male migrant from Eritrea).

This consideration suggests migrants’ intrinsic needs to fully understand the cumulative root causes of mental health burden, from pre-migration trauma to transit time shocks to post-migratory stressors and their influence and the potential exacerbation of other health issues.

Another participant underlined the need for country-specific and individualized approaches to mental health, including the need to avoid over-generalization, which could exacerbate and prevent further health-seeking behavior. The logic behind these considerations could be that, despite their similar post-migration conditions, the stressors, including pre- and during-migration routes, are different. Hence, they might have diverse effects in the migrants’ mental status and stability. As a result, different models of intervention and treatment for mental health conditions might need to be considered and applied separately to different communities. “*It is not good to generalize among refugees, for example among Eritreans and Syrians. As Eritreans, we are different from the Syrians, many arrive with their families together, but we [Eritreans] arrive through challenges of long migration journeys, and mostly we live alone. To solve all those problems, specific treatment procedures need to be adapted to each group, rather than generalizing all together*” (33-year-old male migrant from Eritrea).

### 4.2. PPI on Improving Access to the Healthcare System and Its Utilization

Among the first migrants’ PPI discussions, central themes emerged, such as healthcare system accessibility and its effective utilization and understanding the insurance system. Regarding access, most of the PPI group members agreed that the Swiss healthcare system is not easily understood by refugees, particularly for new arrivals. Despite the availability and legal rights to use it, many of the participants faced difficulties for reaching out and accessing it: “*Even though there are ample healthcare access facilities here in Switzerland, how can we improve our awareness, so that we can utilize the health system provided for us effectively and efficiently?*” (28-year-old female migrant from Eritrea).

A 29-year-old Eritrean female participant highlighted initiatives that were taken by individuals to raise awareness and enhance familiarization with the health system: “*Due to limited awareness, we are not utilizing the system. Initiatives such as those by Dr. Fana Asefaw, are helping women to increase awareness of women’s health*”.

Even after accessing the health system and receiving the service, there is still a sense of dissatisfaction with the diagnosis process and treatment prescription offered. For refugees, the provided diagnosis and treatments seem to be slow and take unnecessary time of their healing and recovering time. Additionally, dosage prescription, antibiotics for example, could be different from the health system in their home countries, as expressed by a 29-year-old Syrian female participant: “*Medically, it’s the best approach to gradually increase the medications’ doses and start with the least ones, but the environment and the culture that we came from make it hard for us to understand that. In Syria, we directly take Augmentin 1000 mg, for example, for a slight flu with fever, and this high consumption of antibiotics did harm us. Now, the lower doses do not work on our bodies at all*”.

Finally, most refugees and asylum seekers in Switzerland that we encountered originated from LMICs with weak healthcare systems and considerable out-of-pocket payments. Hence, they face challenges learning and negotiating the host country’s systems, often based on insurance coverage. This potentially affects how migrants and refugees access healthcare, preventing them from readily integrating and getting early diagnoses and treatment on time, which in turn averts sequelae from complications of treatable diseases. The following were among the issues discussed in the PPI meetings: “*We did not get an explanation as refugees about the type of health insurance we have and the coverage*” (33-year-old male Syrian participant). “*The insurance contracts are too hard to cancel. They need reasons and special dates to be canceled; otherwise, [they] will be renewed automatically. The language barrier is a very important element here*” (26-year-old female Syrian participant). Another participant emphasized that “*Some insurance wages differ from one year to another; we do not understand according to what [or] how to choose the best one when we have the choice*” (27-year-old female Syrian participant).

Particularly participatory health research initiatives among migrants are receiving increasing popularity globally [37,40]. The role of community engagement in research should not be underestimated and can lead to more impactful outcomes if adequately integrated. More rigorous reporting of community engagement is necessary to demonstrate the benefits of participatory health research among forced immigrants and should follow the principles of equity and inclusion in community-academic partnerships [40]. An excellent example is the key role played by the PPI community in raising the level of public involvement in COVID-19 research during the pandemic, as reported by the Health Research Authority in the National Health System (HRA-NHS) in the United Kingdom [35]. Moreover, a co-designed participatory intervention with migrants revealed a successful approach to addressing health inequalities, such as vaccine uptake, within underprivileged communities [41]. Hence, the low public participation rate for vaccine development and clinical trials in LIMCs can be improved through the active engagement of PPI communities following their active contribution [35].

As reflected by our volunteers in this study, PPI can reach beyond their respective communities, and effectively bridge the gap between scientists and the public. However, the “optimal” amount of PPI needs to be estimated for each research project—the aim is to achieve co-creation in a way to leverage migrants’ empowerment to the maximum [37]. As stated by Roura et al., successful participatory initiatives illustrate the value of engaging migrants in co-defining the research agenda, the design and implementation of health interventions, the identification of health-protective factors and the operationalization and validation of indicators to monitor progress [37].

### 4.3. Limitations of the Migration PPI Group Approach

During this study, most of the volunteers were at their initial stage of integration and settlement into the host country. This explains why the mentioned healthcare needs and challenges might seem of more concern than compared to their long-term and more established compatriots. The majority of recently arrived refugees and asylum seekers arrive with poor health conditions [42]. Had it not been for COVID-19, we might have gained a deeper insight due to being able to use more highly effective face-to-face communication and sharing of thoughts rather than digital means. Holding the PPI sessions in person brings documented benefits for an improved understanding of problematic issues and higher efficacy of explaining research aspects to those affected—as reflected by Lampa et al. in the light of COVID-19 [43].

## 5. Conclusions

This small-scale, first-of-its-kind migrants’ PPI initiative in Switzerland highlighted the importance of a participatory approach among migrant communities. It allowed us to effectively identify relevant and problematic issues in migrants, assessing their involvement—or lack thereof—in health-related research and their perceived gaps and concerns in accessing healthcare. The following are conclusions offered for consideration.

First, PPI is a means to increase awareness of migrants to partake in healthcare research-related projects by bridging the scientific and migrant communities and giving them a voice or empowerment to do so. Second, PPI can support healthcare researchers to reach out to migrants and identify important research questions, which are meaningful and ethically accepted. Third, PPI can mitigate migrants’ fears and suspicions by explaining openly the clinical benefits of early screening, diagnosis and treatment options. Fourth, PPI can enhance the understanding of appropriate access and use of the healthcare system of the host country. Fifth, PPI can facilitate health and appropriate the help-seeking behavior of migrants through their PPI members and improve their know-how about health systems and related health-insurance schemes.

## Figures and Tables

**Figure 1 healthcare-12-01654-f001:**
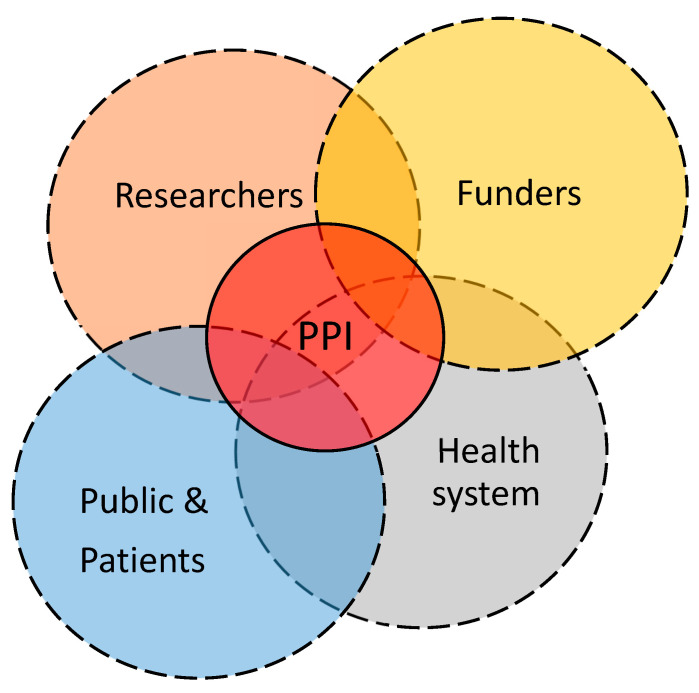
Pivotal role of a public and patient involvement (PPI) strategy in the relationship of key stakeholders of health and care research and services.

**Figure 2 healthcare-12-01654-f002:**
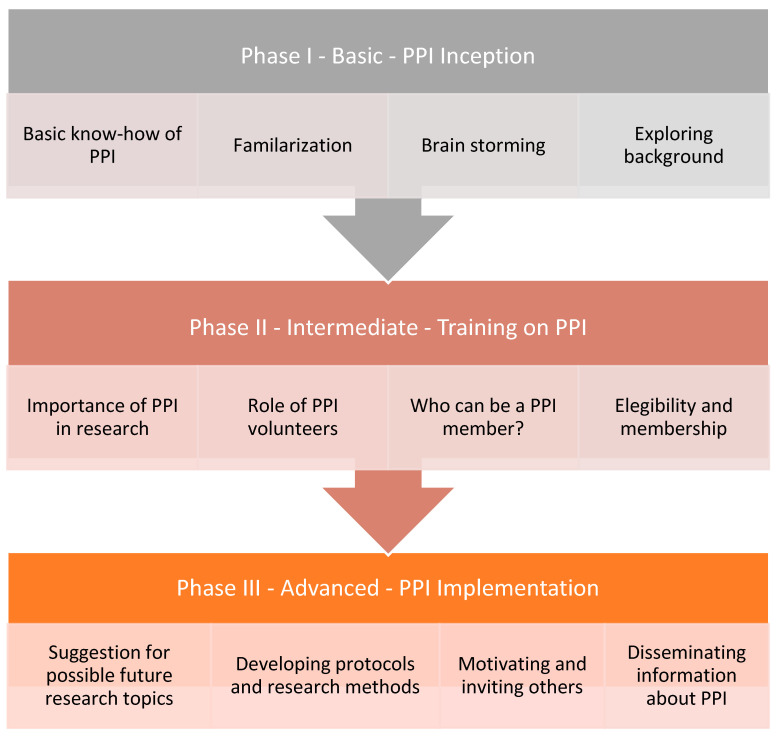
Diagrammatical presentation of establishment and three main phases of public and patient involvement (PPI) for migration health research.

**Table 1 healthcare-12-01654-t001:** Characteristics of participants of a first public and patient involvement (PPI) pertaining to migration health research in Switzerland (N = 14).

Characteristics		Eritrea (N)	Syria (N)	Total N (%)
Gender	Male	5	2	7 (50)
	Female	4	3	7 (50)
Median age (years)		32	27	30.5
Educational attainment	Postgraduate	2	0	2 (14)
	College/graduate	3	4	7 (50)
	High school	4	1	5 (36)
Marital status	Married	9	1	10 (71)
	Single	0	2	2 (14)
	Divorced	0	2	2 (14)
Employment status	Employed	4	2	6 (43)
	Unemployed	5	3	8 (57)
Average duration in Switzerland (years)		7	5	6

## Data Availability

Access to the data can be obtained from the corresponding author upon reasonable request.

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
