# Peer review of "Public and Patient Involvement in Migration Health Research: Eritrean and Syrian Refugees’ and Asylum Seekers’ Views in Switzerland"

_healthcare, 2024, doi:10.3390/healthcare12161654_

Round 1

Reviewer 1 Report

Comments and Suggestions for Authors

I have read this piece of work with keen interest, carefully identifying its merits and weaknesses. I am pleased to declare that the merits far outweigh the minor amendments I will recommend below.

First, the chosen topic significantly contributes to the body of knowledge on refugees' health through a profound bottom-up model, aligning well with the humanitarian values of health promotion and primary healthcare. Second, the manuscript is well-structured and free of language issues. Third, the methodological approach is sound and consistently applied throughout the subsequent analysis.

However, I suggest the authors reconsider the placement of extracts from the interviews in the discussion section. It would be more appropriate to include these in the earlier part of the manuscript.

Additionally, the references list needs some editing; for example, reference 39 seems to be missing parts.

Author Response

I have read this piece of work with keen interest, carefully identifying its merits and weaknesses. I am pleased to declare that the merits far outweigh the minor amendments I will recommend below.

First, the chosen topic significantly contributes to the body of knowledge on refugees' health through a profound bottom-up model, aligning well with the humanitarian values of health promotion and primary healthcare. Second, the manuscript is well-structured and free of language issues. Third, the methodological approach is sound and consistently applied throughout the subsequent analysis.

Response: We are grateful to Reviewer #1 for the overall positive appraisal of our research.

However, I suggest the authors reconsider the placement of extracts from the interviews in the discussion section. It would be more appropriate to include these in the earlier part of the manuscript.

Additionally, the references list needs some editing; for example, reference 39 seems to be missing parts.

Response: We carefully checked all references and amended when need be. In regards to reference #39, there is no missing part of the title (see: https://doi.org/10.1186/s40900-015-0005-8). However, we have abbreviated the journal as “Res Involv Engagem”.

Reviewer 2 Report

Comments and Suggestions for Authors

This is an excellent paper, discussing the findings of an important participatory research project. It points to the centrality of active engagement of refugees/immigrants in all aspects of their settlement and integration in new societies, focusing specifically on healthcare provision and participation. 

The results of this study, although small, if further conceptualised within the context of agency, can have even wider relevance and impact in the field of refugee and migration studies. Hence, an idea for future research and another, related publication.

Author Response

Reviewer #2

This is an excellent paper, discussing the findings of an important participatory research project. It points to the centrality of active engagement of refugees/immigrants in all aspects of their settlement and integration in new societies, focusing specifically on healthcare provision and participation.

The results of this study, although small, if further conceptualized within the context of agency, can have even wider relevance and impact in the field of refugee and migration studies. Hence, an idea for future research and another, related publication.

Response: We thank Reviewer #2 for the generous comments and encouragement to deepen this line of scientific inquiry.

Reviewer 3 Report

Comments and Suggestions for Authors

This paper is important and the research is sound. However, it is not very easy to read and understand. For instance, here are some examples of sentences that are not using proper English grammar and incorrect word choice:

lines 11 and 12: instead of "the role of" it should be "the role that"

line 13: it states "an open and in-depth and interactive consecutive virtual discussion was conducted....." I don't think that an interactive consecutive virtual discussion is an actual thing. I think maybe discussion needs to be plural

line 16: The word "apprehensive" is not used correctly

line 62: this sentence does not make sense (subject-verb agreement)

These are just a few examples of the grammatical problems in the paper. Some of the problems are so great that i can't understand what the authors are trying to say. I think these need to be corrected before this paper is published.

Comments on the Quality of English Language

This paper is important and the research is sound. However, it is not very easy to read and understand. For instance, here are some examples of sentences that are not using proper English grammar and incorrect word choice:

lines 11 and 12: instead of "the role of" it should be "the role that"

line 13: it states "an open and in-depth and interactive consecutive virtual discussion was conducted....." I don't think that an interactive consecutive virtual discussion is an actual thing. I think maybe discussion needs to be plural

line 16: The word "apprehensive" is not used correctly

line 62: this sentence does not make sense (subject-verb agreement)

These are just a few examples of the grammatical problems in the paper. Some of the problems are so great that i can't understand what the authors are trying to say. I think these need to be corrected before this paper is published.

Author Response

Reviewer #3

This paper is important and the research is sound. However, it is not very easy to read and understand. For instance, here are some examples of sentences that are not using proper English grammar and incorrect word choice:

= lines 11 and 12: instead of "the role of" it should be "the role that"

= line 13: it states "an open and in-depth and interactive consecutive virtual discussion was conducted....." I don't think that an interactive consecutive virtual discussion is an actual thing. I think maybe discussion needs to be plural

= line 16: The word "apprehensive" is not used correctly

= line 62: this sentence does not make sense (subject-verb agreement)

These are just a few examples of the grammatical problems in the paper. Some of the problems are so great that i can't understand what the authors are trying to say. I think these need to be corrected before this paper is published.

Response: We thank Reviewer #3 for scrutinizing our piece and providing specific examples where careful rewording is required to enhance clarity and comprehension. Hence, we thoroughly reworked our piece, placing particular emphasis on grammar, punctuation, and style. Specifically, the four issues highlighted have been addressed as follows;

  • Lines 11-12: Corrected as suggested (see the revised manuscript, final clean version, line 12)
  • Line 13: The plurality of the ‘discussion was’ is re-considered as ‘discussions’ ((see the revised manuscript, final clean version, line 14)
  • Line 16: The underlined word is replaced by the phrase ‘was not straightforward’ ((see revised manuscript, final clean version, line 16)
  • Line 62: The grammatical concern has been addressed (see revised manuscript, final clean version, lines 62-63)